# Common Trends and Differences in Antioxidant Activity Analysis of Phenolic Substances Using Single Electron Transfer Based Assays

**DOI:** 10.3390/molecules26051244

**Published:** 2021-02-25

**Authors:** Melanie Platzer, Sandra Kiese, Thomas Herfellner, Ute Schweiggert-Weisz, Oliver Miesbauer, Peter Eisner

**Affiliations:** 1ZIEL-Institute for Food & Health, TUM School of Life Sciences Weihenstephan, Technical University of Munich, Weihenstephaner Berg 1, 85354 Freising, Germany; peter.eisner@ivv.fraunhofer.de; 2Fraunhofer Institue for Process Engineering and Packaging IVV, Giggenhauser Str. 35, 85354 Freising, Germany; sandra.kiese@ivv.fraunhofer.de (S.K.); thomas.herfellner@ivv.fraunhofer.de (T.H.); ute.weisz@ivv.fraunhofer.de (U.S.-W.); 3Chair of Food Packaging Technology, TUM School of Life Sciences Weihenstephan, Technical University of Munich, Weihenstephaner Steig 22, 85354 Freising, Germany; oliver.miesbauer@tum.de; 4Chair of Food Science, Institute for Nutritional and Food Sciences, University of Bonn, Meckenheimer Allee 166a, 53113 Bonn, Germany

**Keywords:** antioxidant capacity, flavonoids, phenolic acids, reaction kinetics, stoichiometry

## Abstract

Numerous assays were developed to measure the antioxidant activity, but each has limitations and the results obtained by different methods are not always comparable. Popular examples are the DPPH and ABTS assay. Our aim was to study similarities and differences of these two assay regarding the measured antioxidant potentials of 24 phenolic compounds using the same measurement and evaluation methods. This should allow conclusions to be drawn as to whether one of the assays is more suitable for measuring specific subgroups like phenolic acids, flavonols, flavanones, dihydrochalcones or flavanols. The assays showed common trends for the mean values of most of the subgroups. Some dihydrochalcones and flavanones did not react with the DPPH radical in contrast to the ABTS radical, leading to significant differences. Therefore, to determine the antioxidant potential of dihydrochalcone or flavanone-rich extracts, the ABTS assay should be preferred. We found that the results of the flavonoids in the DPPH assay were dependent on the Bors criteria, whereas the structure–activity relationship in the ABTS assay was not clear. For the phenolic acids, the results in the ABTS assay were only high for pyrogallol structures, while the DPPH assay was mainly determined by the number of OH groups.

## 1. Introduction

Plant-based antioxidants are known for their ability to limit radical reactions by transferring hydrogen atoms or electrons and to interrupt the chain reactions of oxidative degradation [1,2,3,4,5]. Among the most important groups of plant-based antioxidants are phenolic compounds, which have one or more aromatic rings with one or more hydroxy groups. They are common plant secondary metabolites and are divided into several major families according to their chemical structure, including the flavonoids and phenolic acids [6]. The phenolic compounds investigated in this article are listed in Table 1.

Three criteria for the structure-activity relationship (SAR) of antioxidant compounds have been defined by Bors [7]:Bors 1—due to hydrogen bonding, the presence of a catechol group on the B-ring leads to a high stability of the antioxidant radical (AO·);Bors 2—a 2,3 double bond in combination with a 4-oxo group on the C-ring facilitates electron delocalization;Bors 3—the presence of OH groups at position 3 and 5 in combination with the 4-oxo group enables electron delocalization via hydrogen bonds.

The presence of a pyrogallol structure on the B-ring can also increase the antioxidant activity [8]. Molecules that fulfil all of the Bors’ criteria, such as quercetin (QUR) and myricetin (MYR), should therefore achieve the most efficient electron delocalization and accordingly should possess the highest antioxidant activity [7,9]. Attempts to correlate chemical structures with antioxidant activity are usually based on the analysis of natural phenolic compounds and extracts using various antioxidant assays [3]. The most widely used methods are based on oxygen radical absorbance capacity (ORAC) [10], 2,2${}^{\prime }$-azino-bis (3-ethylbenzothiazoline-6-sulfonic acid) (ABTS) [11] and 2,2-diphenyl-1-picrylhydrazyl (DPPH) [12] assays. The underlying chemistry involves either hydrogen atom transfer (HAT) or single electron transfer (SET) as shown in Figure 1 [3].

HAT assays (such as ORAC) measure the ability of an antioxidant (AOH) to inactivate a free radical (ROO·) by releasing a hydrogen atom [3,4,14]. In contrast, assays that are dominated by SET-based reaction mechanisms, such as ABTS and DPPH, measure the release of an electron to the (ROO·), converting it into an anion (ROO^−^) [4,15,16]. The latter causes reversible deprotonation and a color decrease in the solution, which not only indicates the reaction endpoint but simultaneously reports the concentration of the antioxidant [3,17]. SET mechanisms can be subdivided into SET-PT and SPLET. The two-stage SET-PT mechanism strongly depends on the ionization potential and proton dissociation energy. Antioxidants, which are easily ionized and deprotonated, are highly reactive. In contrast, the SPLET mechanism involves the initial loss of a proton from the antioxidant followed by anion transfer to the radical, which then reacts with the proton. This is influenced by the proton affinity and electron transfer enthalpy. SET-PT and SPLET are in thermodynamic equilibrium with each other (and also with the HAT mechanism), because the reactants and end products are identical. However, the reaction rate in each assay depends on different physical processes [4,15,16]. The dominant reaction depends on the pH and solvent [16], resulting in different SARs for each assay. The ABTS radical preferably reacts via the SPLET mechanism in aqueous solutions, whereas the DPPH radical preferably reacts via the SPLET mechanism in solvents such as ethanol and methanol [18,19]. Some authors have theoretically investigated the reaction mechanisms and classified the antioxidant effect of substances via the SPLET mechanism on the basis of electron transfer energy and proton affinity, but the dependence of these effects on the pH and solvent makes it difficult to make accurate predictions [20,21,22].

Each of these assays has unique properties and limitations, making it difficult to compare results generated using different methods—for example, measurement at a fixed time or kinetic, measurement of different concentrations or only one concentration, and the use of different solvents [23,24,25,26]. Furthermore, the outcome of the assays can be evaluated in different ways, resulting in metrics, such as half maximal effective concentration (EC_50_), time to reach the EC_50_ (T_EC_50__), antiradical efficiency, antiradical power (ARP), stoichiometry, kinetic behavior and rate constant, radical scavenging activity, or trolox equivalent antioxidant capacity (TEAC) [23,24,27,28,29,30,31,32]. Although there is a large number of publications on the assays and their limitations, little is found on the most suitable use of the assays for different types and applications of extracts.

Our aim was to study whether the DPPH or the ABTS assay should be preferred to analyze the antioxidant potential of extracts rich in different phenolic components, or whether they lead to similar results. Therefore, we investigated the SAR of 24 different phenolic compounds by comparing two SET assays (ABTS and DPPH). The phenolic compounds represented the phenolic acids, flavonols, flavanones, dihydrochalcones, and flavanols, allowing us to compare the antioxidant properties of different groups as well as individual substances within a group and thus determine structural features correlating with high antioxidant activity. We used the same measurement as well as evaluation methods in our study. To ensure that the reactions were complete and the reported values are not underestimated, all measurements were recorded as a function of time. Furthermore, second order kinetic equations were used to verify the endpoint values. EC_50_, ARP and stoichiometry values are interconvertible without affecting the outcome, so all results are presented herein as stoichiometry values [29].

## 2. Materials and Methods

Chemicals (antioxidant reference standards) were obtained from MilliporeSigma (Steinheim, Germany): caffeic acid (CAA), (+)-catechin (CAT), 3,4-dihydroxybenzoic acid (DBA), (−)-epicatechin (EPC), ferulic acid (FEA), gallic acid (GAA), 4-hydroxybenzoic acid (HBA), hesperetin (HES), kaempferol (KAE), morin (MOR), myricetin (MYR), naringenin (NAN), p-coumaric acid (PCA), proanthocyanidin B1 (PCB1), proanthocyanidin B2 (PCB2), phloridzin (PHD), phloretin (PHT), quercetin-3-D-galactosides (QGA3), quercetin-3-D-glucosides (QGU3), quercetin QUR, sinapic acid (SIA), siringic acid (SRA) and taxifolin (TAF). These are shown in Table 1. We also obtained DPPH radical, ABTS diammonium salt and potassium persulfate from the same supplier. The narirutin (NAR) reference standard was purchased from J&K Scientific (Marbach am Neckar, Germany) and is also shown in Table 1. Stock solutions were prepared by dissolving the reference standards in absolute ethanol (analytical grade), taking into account the purity, and diluting them in seven steps (0.075–1 mM) for the measurements.

The antioxidant activity was measured using the DPPH [23] and ABTS [11] methods as previously described, with slight modifications. The stock solution for the DPPH assay was prepared by dissolving 24 mg of the radical in 100 mL absolute ethanol. The working solution was prepared diluting the stock solution 1:10. We mixed 50 μL of the sample (reference standard dilution or ethanol blank) with 1950 μL of the working solution for each measurement, and the absorbance was determined by spectrophotometry at 515 nm [23]. The stock solution for the ABTS assay was prepared by dissolving 6.62 mg potassium persulfate and 38.4 μg ABTS diammonium salt in 10 mL demineralized water. This solution was incubated in the dark for 12–16 h and then the working solution was prepared by diluting 1:100 with demineralized water. We mixed 10 μL of each sample (reference standard dilution or ethanol blank) with 990 μL of the working solution for each measurement, and the absorbance was determined by spectrophotometry at 734 nm using a Specord 210 plus spectrophotometer (Analytik Jena, Jena, Germany) [11]. In both assays the absorption was measured as a function of time and the color decrease of the respective radical (DPPH or ABTS) was detected. All measurements were performed in triplicate. The decrease of absorbance in percent was then used to calculate the decrease of initial radical concentration in percent. This corresponds to the amount of radical reduced in mM and was plotted as a function of time. The reaction mechanism is explained below using the DPPH radical, but the principle is the same in the ABTS assay. As mentioned above, all HAT- and SET-based mechanisms lead to identical products (see Figure 1) and therefore can be summarized to Equation (1), where the AOH reacts with DPPH· to form the intermediate AO·
(1)AOH+αDPPH·→DPPH−H+AO·,
which reacts with DPPH· in a second step to yield DPPH-AO Equation (2)
(2)AO·+βDPPH·→DPPH−AO,
where α and β are stoichiometric coefficients [23,29]. Since only the color decrease of the DPPH radical (DPPH·) is measured and it is not possible to distinguish in which of the Equations (1) or (2) the radical reacts, only a total stoichiometry (α + β) can be determined with this method. Furthermore, as shown in Equation (3): (3)AO·+AO·→AO−AO.

Since this reaction does not cause any color decrease in the assay, it is neglected for the following considerations. The reaction of an antioxidant with the DPPH and ABTS reagents follows second-order kinetics [28,29]. Therefore, the changing concentrations of AOH, AO· and DPPH· over time are given as shown in Equations (4)–(6): (4)$$\begin{array}{cc} \frac{d\left[\mathrm{AOH}\right]}{dt} & =-k_{1}\left[\mathrm{AOH}\right]\left[\mathrm{DPPH}\cdot \right], \end{array}$$(5)$$\begin{array}{cc} \frac{d\left[\mathrm{AO}\cdot \right]}{dt} & =k_{1}\left[\mathrm{AOH}\right]\left[\mathrm{DPPH}\cdot \right]-k_{2}\left[\mathrm{AO}\cdot \right]\left[\mathrm{DPPH}\cdot \right], \end{array}$$(6)$$\begin{array}{cc} \frac{d\left[\mathrm{DPPH}\cdot \right]}{dt} & =-\alpha k_{1}\left[\mathrm{AOH}\right]\left[\mathrm{DPPH}\cdot \right]-\beta k_{2}\left[\mathrm{AO}\cdot \right]\left[\mathrm{DPPH}\cdot \right], \end{array}$$
where $k_{1}$ and $k_{2}$ are the reaction rate constants. The initial concentrations ${\left[\mathrm{AOH}\right]}_{0}$ and ${\left[\mathrm{DPPH}\cdot \right]}_{0}$ are assumed to be positive, whereas ${\left[\mathrm{DPPH}-H\right]}_{0}$ and ${\left[\mathrm{AO}\cdot \right]}_{0}$ are set to zero. In our experiments, DPPH· was available in excess and accordingly, we assume that its concentration $\left[\mathrm{DPPH}\cdot \right]$ on the right side of Equations (4)–(6) is constant. In this case, the reactions follow pseudo-first order kinetics [33] as shown in Equations (7)–(9): (7)$$\begin{array}{cc} \frac{d\left[\mathrm{AOH}\right]}{dt} & =-{\tilde{k}}_{1}\left[\mathrm{AOH}\right], \end{array}$$(8)$$\begin{array}{cc} \frac{d\left[\mathrm{AO}\cdot \right]}{dt} & ={\tilde{k}}_{1}\left[\mathrm{AOH}\right]-{\tilde{k}}_{2}\left[\mathrm{AO}\cdot \right], \end{array}$$(9)$$\begin{array}{cc} \frac{d\left[\mathrm{DPPH}\cdot \right]}{dt} & =-\alpha {\tilde{k}}_{1}\left[\mathrm{AOH}\right]-\beta {\tilde{k}}_{2}\left[\mathrm{AO}\cdot \right], \end{array}$$
where the reaction rate constants are defined as ${\tilde{k}}_{1}=k_{1}{\left[\mathrm{DPPH}\cdot \right]}_{0}$ and ${\tilde{k}}_{2}=k_{2}{\left[\mathrm{DPPH}\cdot \right]}_{0}$. Equation (7) is a first-order linear homogenous differential equation and its analytical solution is given by
(10)$${\left[\mathrm{AOH}\right]}_{t}={\left[\mathrm{AOH}\right]}_{0}\exp\left(-{\tilde{k}}_{1}t\right).$$

This expression is substituted in Equation (8), which is a first-order linear inhomogeneous differential equation. Its solution is obtained as the sum of the general solution of the corresponding homogeneous equation and a particular solution of the inhomogeneous equation, which is determined by an exponential ansatz as well. Considering the initial condition ${\left[\mathrm{AO}\cdot \right]}_{0}=0$ gives
(11)$${\left[\mathrm{AO}\cdot \right]}_{t}=\frac{{\left[\mathrm{AOH}\right]}_{0}{\tilde{k}}_{1}}{{\tilde{k}}_{2}-{\tilde{k}}_{1}}\left(\exp\left(-{\tilde{k}}_{1}t\right)-\exp\left(-{\tilde{k}}_{2}t\right)\right).$$

Substituting ${\left[\mathrm{AOH}\right]}_{t}$ and ${\left[\mathrm{AO}\cdot \right]}_{t}$ in Equation (9) and integrating over time leads to the expression for ${\left[\mathrm{DPPH}\right]}_{t}$, allowing to calculate the amount of DPPH· per volume, which is consumed up to time *t*, as shown in Equation (12):(12)$$\begin{array}{cc} \Delta {\left[\mathrm{DPPH}\cdot \right]}_{t} & ={\left[\mathrm{DPPH}\cdot \right]}_{0}-{\left[\mathrm{DPPH}\cdot \right]}_{t}\\ \; & ={\left[\mathrm{AOH}\right]}_{0}\left(\alpha +\beta +\frac{\alpha {\tilde{k}}_{1}-\left(\alpha +\beta \right){\tilde{k}}_{2}}{{\tilde{k}}_{2}-{\tilde{k}}_{1}}\exp\left(-{\tilde{k}}_{1}t\right)+\frac{\beta {\tilde{k}}_{1}}{{\tilde{k}}_{2}-{\tilde{k}}_{1}}\exp\left(-{\tilde{k}}_{2}t\right)\right). \end{array}$$

The steady-state values are the asymptotic values of Equation (12) for t→∞. We adapted this model function to the concentration of reduced radicals, allowing the stationary endpoint of the kinetics to be determined even if the reaction did not reach completion within the measurement time. Assuming that the AOH had completely reacted with the radicals by the reaction endpoint, the measured amount of reduced DPPH and ABTS was plotted as a function of the AOH concentration initially used (${\left[\mathrm{AOH}\right]}_{0}$). When plotting the concentration of reduced DPPH ($\Delta {\left[\mathrm{DPPH}\cdot \right]}_{t}={\left[\mathrm{DPPH}\cdot \right]}_{0}-{\left[\mathrm{DPPH}\cdot \right]}_{t}$), the positive slope of the linear regression directly indicates the total stoichiometry of the number of DPPH radicals, needed to oxidize the complete AOH in all subreactions (steady state). As shown in Figure 2 as a schematic example, the last two points are in the saturation range, since all DPPH radicals have already been consumed here. In order not to underestimate the stoichiometry, only the concentrations below this range were used for the linear regression. Furthermore, the linear regression equation can be used to calculate the amount of reduced DPPH for higher AOH concentrations, which cannot be determined experimentally.

To determine the stoichiometry values after 5 and 30 min, the fit values after 5 and 30 min were evaluated in the same manner as the steady state stoichiometry. The kinetic behavior of each substance was assigned to one of three groups (fast, medium and slow) as previously described [23]. The fast substances reached steady state within 5 min and the medium ones within 30 min. All substances that needed more than 30 min to reach steady state were assigned to the slow kinetic behavior group. The slower the reaction rate, the more complex the reaction, as previously shown using butylated hydroxytoluene [30].

Statistical evaluation was carried out by one-way analysis of variance (ANOVA) with all significant decimal places using Sigma Plot (Systat Software, San Jose, CA, USA), corresponding to an unpaired *t*-test. If there was a significant difference, an additional pairwise test was performed using the Holm–Šidák method. The significance level for both tests was 0.05.

## 3. Results and Discussion

The antioxidant activities of 24 polyphenolic reference standards representing five different groups of compounds were determined using DPPH and ABTS assays. The concentration of reduced radicals was then plotted as a function of time. TAF, DBA and CAA are compared in Figure 3 as examples of slow, medium and fast kinetics, showing that only CAA reached the final value after 20 min.

If the value of DBA or TAF is measured after 20 min in the absence of kinetic data, the true value can therefore be underestimated, and in some cases a different kinetic order for the antioxidant effect might be predicted. Accordingly, we checked whether the extrapolated final values differed significantly when the measured values at 13, 17 and 20 min were used to adjust Equation (12). If there was a significant difference between the values predicted at these time points, the assay was repeated with longer measurement durations. Figure 4 shows the value reported for EPC after 14 h.

Here, there was no significant difference between the calculated final values after adapting Equation (12) using the values measured at 11 and 14 h so the measurement duration was considered sufficient. The substances that required longer measurement durations during the DPPH assay and ABTS assay are discussed in more detail below. In the following, we report the results for the subgroups using boxplots for the DPPH (Figure 5) and ABTS (Figure 6) assays and the results of all used phenolic compounds in both assays (Figure 7).

### 3.1. Antioxidant Activity Determined Using the DPPH Test

The antioxidant activities based on the DPPH assay are shown as stoichiometry boxplots in Figure 5. The sequence of decreasing mean values was: flavanol oligomers > flavanol monomers > flavonols > phenolic acids > flavanones > dihydrochalcones, which is largely consistent with previous reports with the exception of the dihydrochalcones [34]. The order of the flavonoid subgroups mainly depended on the number and position of OH groups, which is why the flavanol oligomers (PCB1 and PCB2) consisting of two flavanol monomers (EPC and CAT) achieved a higher mean value [34]. In addition, the Bors criteria appeared to play a decisive role [7]. Accordingly, the flavanols display high antioxidant values because they fulfill Bors criterion 1, which seems to be the most important of the three [35]. Similarly, the flavonols fulfill Bors criterion 2 and most of them fulfill Bors criterion 3 and also carry a catechin or pyrogallol group on the B-ring. Phenolic acids are difficult to compare with flavonoids because these groups are structurally distinct, but we observed a wide range of antioxidant values in part reflecting the high values assigned to hydroxybenzoic acids but the lower values assigned to hydroxycinnamic acids. The low mean value of the flavanones reflects the low number of OH groups and the two compounds (NAR and NAN) that did not react with the DPPH radical. The dihydrochalcones had the lowest values overall, reflecting the low number of OH groups and the open C-ring of PHT. Furthermore, PHD did not react with the DPPH radical in our assay. Because most of the compounds fulfill Bors criterion 3, it is not possible to make a definitive statement concerning its influence on the SAR.

The individual results of all reference standards in the DPPH assay are shown in Figure 7a and depend mainly on Bors criterion 1. All substances with a catechol or pyrogallol group on the B-ring showed high antioxidant values. In contrast to Bors’ statement that substances meeting all three criteria are the most potent antioxidants [7], MYR and QUR achieved only medium-high activity and the most active compounds in our DPPH assay fulfilled only Bors criterion 1. HBA, FEA, PCA, SIA and SRA each possess only one OH group, and these showed low antioxidant values. An OH group in the *para* position confers more activity than one in the *meta* position, which is why HES also showed low activity [23,34,36,37,38,39,40,41]. The references NAR, NAN and PHD did not react with the DPPH radical, perhaps due to steric hindrance. The stoichiometry and kinetic behavior of all reference standards are summarized in Table 2.

For the **phenolic acids**, the ranking of antioxidant activity was GAA ≫ DBA > SRA ≈ CAA > SIA > PCA > FEA ≫ HBA in broad agreement with the literature [34,41,47,48]. In the case of PCA, the extrapolated final values differed significantly after adjusting the curve to 1000 and 1200 s using Equation (12), so we extended the duration of measurement. The results depended mainly on the number of OH groups on the aromatic ring, which is why GAA with three groups was more active than DBA and CAA, which both possess a catechol group. Substances with only one OH group had the weakest antioxidant activity (SIA, PCA, FEA and HBA). GAA showed the highest values due to its pyrogallol-like structure and ability to release hydrogen atoms, which is particularly important for hydrogen transfer [34,40]. As stated above, OH groups at the *para* position confer high antioxidant activity [23,34,36,37,38,39,40,41]. We found that hydroxybenzoic acids (GAA, DBA and SRA) were more potent than hydroxycinnamic acids (CAA, SIA, FEA and PCA), also in agreement with the literature [34,49,50]. Hydroxycinnamic acid could play a major role in the stabilization of the radical by resonance and also has a large hydrogen release capacity, leading to the stabilization of the resulting radical [34]. The order for the hydroxycinnamic acids we tested was SRA > SIA > FEA > PCA, suggesting that the methoxy group also influences stoichiometry in the DPPH assay, as previously reported [34,40]. The methoxy group is an electron donor, reducing the bond dissociation energy and therefore promoting electron transfer [13].

GAA, CAA and HBA were the phenolic acids with the fastest kinetics, partly in agreement with the literature [42,51,52]. The presence of a catechol, pyrogallol or single OH group appeared to have little effect on reaction kinetics. Phenolic acids in the medium group (SRA, SIA, DBA and FEA) did not react completely within 30 min, which may lead to an underestimation of their antioxidant activity. FEA was shown to have medium kinetics in a previous study [29]. PCA was the only phenolic acid with a slow kinetics in our assay.

For the **flavonols**, the ranking of antioxidant activity was MYR > QGU3 > QGA3 > QUR > KAE > MOR, when the compounds were dissolved in ethanol. Another study reported a similar order to that observed in our experiments (MYR > QUR > MOR = KAE) [27]. Because flavonols fulfill Bors criterion 2 and in many cases also criterion 3, we do not discuss them further here. The highest activity was observed for substances with an additional catechol group on the B-ring, which also corresponds to the Bors criteria. MYR has catechol and OH groups, thus explaining its slightly higher value than QUR (with a lone catechol group) but indicating that the additional OH group has only a small influence. The presence of a sugar residue likewise has only a small influence, explaining why the two glycosylated quercetins showed similar values to the quercetin aglycone. In methanol, the sugar residues were found to slightly inhibit antioxidant activity [35]. KAE and MOR have no catechol group and therefore showed the lowest values in our assay. All substances showed medium kinetics except KAE, which was assigned to the fast kinetics group. In contrast to the phenolic acids, the presence of catechol or pyrogallol groups did appear to affect the reaction velocity, explaining why the quercetin derivatives and MYR behaved in the same manner. All substances assigned to the medium kinetics category featured two OH groups on the B-ring.

For the **flavanols**, the ranking of antioxidant activity was PCB1 > CAT > EPC ≈ PCB2. EPC and CAT are structural isomers and were measured over a longer duration because data from the 30-min assay were insufficient. EPC and CAT produced similar yet significantly different activities. PCB1 and PCB2 are also structural isomers, but PCB2 showed a much lower antioxidant activity probably due to steric hindrance. PCB2 was assigned to the medium kinetics category, whereas PCB1, EPC and CAT were assigned to the slow kinetics category.

For the **flavanones**, only two of the four standard references reacted with the DPPH radicals, with TAF showing much greater activity than HES. NAR and NAN are structurally similar, but NAR carries an additional sugar residue at position 7 on the A-ring, which could prevent interaction with DPPH due to steric hindrance. The minimal activity of HES requires further investigation. TAF fulfills Bors criteria 1 and 3, thus achieving a higher value than HES. Furthermore, HES features only one OH group (in the *meta* position of the B-ring) in contrast to the catechol group of TAF. Both TAF and HES showed slow kinetic behavior.

Finally, we tested two **dihydrochalcones** (PHD and PHT) only the latter of which reacted with DPPH. PHD and PHT are structurally similar, differing only in the presence of a sugar residue at position 6’ on the A-ring of PHD, which as stated above for NAR could prevent interaction with the DPPH radical due to steric hindrance. PHT showed a low reaction stoichiometry and slow kinetic behavior. PHD has been reported to react moderately or slowly with DPPH, but these experiments involved different solvents [42,53].

In summary, our DPPH assay results partly agree with the literature, with discrepancies likely to reflect the different solvents used and other differences in the measurement and/or evaluation methods. Our results showed no clear correlation between the structure of the polyphenolic molecules and the reaction rate. The intermediate product AO· cannot be distinguished from the final product by spectrophotometry, hence the speeds of the two reactions and the reaction rates ${\tilde{k}}_{1}$ and ${\tilde{k}}_{2}$ cannot be determined individually. The assignment of reaction speeds [23] therefore indicates an average value for both reactions. To achieve a high value in the assay, the three Bors criteria and the number of OH groups on the molecule appear decisive. Furthermore, the presence of an OH group in the *para* position of the B-ring is important and the presence of a sugar residue has a positive effect, regardless of the type of sugar [23,34,36,37,38,39,40,41].

### 3.2. Antioxidant Activity Determined Using the ABTS Test

The antioxidant activities based on the ABTS assay are shown as stoichiometry boxplots in Figure 6. The sequence of decreasing mean values was: flavanol oligomers > dihydrochalcones > flavanol monomers > flavonols > phenolic acids > flavanones, which again is largely consistent with the literature with the exception of the dihydrochalcones [34]. In general, the number of OH groups seemed to determine the order. The high values of dihydrochalcones may reflect the lower steric hindrance of their relatively open structures. When comparing all 24 substances regardless of their subgroup, there were no conspicuous structural features that explained the ranking [37]. The stoichiometry and kinetic behavior of all the reference standards are shown in Table 3. However, ABTS results from literature are not shown because they are usually provided as TEAC values, which correlate poorly with the reaction kinetics [54]. The individual results of all reference standards in the ABTS assay are shown in Figure 7b.

For the **phenolic acids**, the ranking of antioxidant activity was GAA ≫ FEA > SIA ≈ PCA > CAA > SRA > DBA ≈ HBA. GAA achieved the highest value due to the pyrogallol group on the aromatic ring. However, the presence of a single OH or catechol group appeared to be less important, explaining the similar values of HBA vs DBA and CAA vs PCA, all lower than GAA. Furthermore, the hydroxybenzoic acids appeared more active than the hydroxycinnamic acids, the exception being the benzoic acid GAA, which achieved the highest value. The presence of a methoxy group also affected the activity, explaining why SRA, SIA and FEA achieved higher values than HBA and PCA. FEA, which has two methoxy groups, achieved a higher value than SIA with only one. CAA, FEA, GAA and SIA were assigned to the medium kinetics category, whereas all the others showed fast kinetics. There was no clear correlation between the reaction velocity and chemical structure, and the type of acid group appeared to have no influence on the velocity of the reaction.

For the **flavonols**, the ranking of antioxidant activity was QUR > KAE > MOR > MYR > QGU3 > QGA3. As discussed for the phenolic acids, the activity of the flavonols does not appear to depend on the presence of the catechol group, which is why QUR and KAE reached similar values. Additional OH groups had a negative impact, hence MYR achieved a much lower value than QUR, possibly caused by steric hindrance. Furthermore, the presence of any sugar residue also caused a negative impact. A second OH group on the B-ring only influenced the activity if it formed a catechol group, which is why MOR did not show a higher value than KAE. The flavonol standard references showed medium kinetics, except QUR and KAE, which were assigned to the slow kinetics category. For these two substances, it was necessary to repeat the measurement with an extended duration, although it is unclear which chemical groups were responsible for the effect. There may be steric hindrance in some molecules, for example due to the presence of a sugar residue or a pyrogallol group.

For the **flavanols**, the ranking of antioxidant activity was PCB1 > PCB2 > EPC ≈ CAT. As stated above, EPC and CAT are structural isomers and so are PCB1 and PCB2. PCB2 consists of two EPC molecules and thus has twice the number of catechol and OH groups, explaining its higher activity than EPC/CAT. PCB1 achieved the highest value overall, suggesting that PCB2 again suffers from a steric hindrance effect. All standard references showed medium kinetic behavior, which was expected due to the structural similarity of the molecules.

For the **flavanones**, the ranking of antioxidant activity was NAN > NAR > TAF > HES. The order appeared to depend mainly on the position of the OH group in the B-ring. HES, with its *meta* OH group, showed the lowest activity, whereas the more active TAF, NAR and NAN all have an OH group in the *para* position [23,34,36,37,38,39,40,41]. The presence of a sugar residue appeared to inhibit antioxidant activity, hence the lower value of NAR compared to NAN. TAF achieved a higher value than NAN because it fulfills the second and third Bors criteria. NAR and NAN were assigned to the slow kinetics category requiring a longer duration of measurement, whereas TAF and HES showed fast kinetics.

Finally, we found that both **dihydrochalcones** (PHD and PHT) were active in the ABTS assay, although PHT was more active than PHD. The only difference between these compounds is the presence of a sugar residue on the A-ring of PHD, which appears to exert a negative influence on antioxidant activity. Both PHD and PHT showed slow kinetic behavior and were analyzed by extending the measurement duration.

In summary, it was not possible to compare the sequences of all substances because different trends were observed within each group, hence no SAR and no clear correlation between molecular structure and reaction rate in the assay could be established. The precise structural properties that are important in the ABTS assay could not be determined, and the Bors criteria seem to play a minor role. The kinetic behavior of the reaction reflects the complexity of the reactant, with large polyphenolic molecules reacting more slowly than simpler ones because the former must reorient before reaction with the ABTS radical. However, substances that show fast kinetics can also have low antioxidant capacities, and substances that show a high stoichiometry with the ABTS radical do not necessarily achieve high reactivity [54].

### 3.3. Comparison of the DPPH and ABTS Assays

When comparing the boxplots (Figure 5 and Figure 6), the mean values of the polyphenol subgroups, with the exception of the dihydrochalcones, revealed similar trends in both assays. The subgroup of the dihydrochalcones showed radically different mean values, reflecting the fact that one of the two tested compounds did not react with the DPPH radical.

We observed no clear correlation when comparing the results of the individual substances in the two assays, suggesting that antioxidant activity is dependent on multiple criteria (Figure 7).

Results in the DPPH assay appeared to depend mainly on the number of OH groups and Bors criteria 1 and 3, but further investigation is required because most of the standard reference compounds we used conformed at least to Bors criterion 3. In contrast, there was no clear relationship between the results in the ABTS assay and the number of OH groups, and the Bors criteria were much less important.

For the **phenolic acids**, the hydroxycinnamic acids achieved higher values than the hydroxybenzoic acids in the ABTS assay, whereas this was not the case in the DPPH assay. Furthermore, compounds with an additional methoxy group showed higher activities in both assays. However, compounds with both OH and methoxy groups achieved higher values than those with a catechol group in the ABTS assay but not in the DPPH assay. For the DPPH assay, activity was mainly dependent on the number of OH groups and the presence of catechol or pyrogallol groups.

For the **flavonols**, the first Bors criterion was important in both assays. An additional OH group on the B-ring increased the activity in the DPPH assay but not the ABTS assay. The presence of a sugar residue conferred a slightly negative effect in the ABTS assay but had no significant impact on the DPPH assay. The presence of an OH group at position 2 did not affect the results of either assay.

For the **flavanones**, the presence of an OH group at position 3 in the C-ring had a major effect on the results of the DPPH assay, and the presence of a sugar residue tended to abolish the reaction. The *para* position of an OH group in the B-ring appeared to play an important role in both assays.

Among the four **flavanols** we tested, PCB1 achieved high values in both assays, perhaps due to the high number of catechol and OH groups. In contrast, PCB2 showed high activity only in the ABTS assay, suggesting steric hindrance may inhibit its activity in the DPPH assay. CAT and EPC are structural isomers that do not seem prone to steric hindrance, thus the differences between these compounds were small in both assays.

Finally, although we tested only two **dihydrochalcones**, making it difficult to draw general conclusions for this group of molecules, it appears that the sugar residue on PHD had a negative impact on activity, and this aspect could be explored by testing more diverse compounds.

## 4. Conclusions

The mean values of most of the polyphenol subgroups revealed similar trends in both assays. The dihydrochalcones were the only compounds to show radically different mean values, reflecting the fact that we tested only two compounds and one of them did not react with the DPPH radical. Therefore, dihydrochalcone-rich extracts should not be measured using the DPPH assay because this underestimates the antioxidant activity. The same caveat applies to the flavanones. If these substances are divided into subgroups, the assays reported different results despite the identical measurement and evaluation methods. In general, the results of the DPPH assay correlated mainly with the Bors criteria, whereas the SAR was not clear in the ABTS assay. This may be due the type of solvent, which was not possible to be the same in the two assays. Furthermore, the DPPH and ABTS model radicals are structurally distinct. In terms of kinetic behavior, we observed no clear correlation between structure and reaction velocity. However, to ensure that endpoint values are reported, the duration of measurement should be at least 30 min and Equation (12) should be used to calculate the final values, thus avoiding measurement times of several hours. This also ensures that the measurement duration can be extended if necessary. Additionally, the potential synergistic effects of pure substances should be taken into account when measuring the reaction kinetics of extracts containing mixtures of substances that react at different speeds. In order to use these assays as rapid tests for specific applications, they must be matched to the antioxidant effect required in the application medium (e.g., food matrix or body tissue). Other relevant parameters such as pH, solvent, and the content of carbohydrate and protein should also be considered, because these may affect the reported activity of natural extracts.

## Figures and Tables

**Figure 1 molecules-26-01244-f001:**
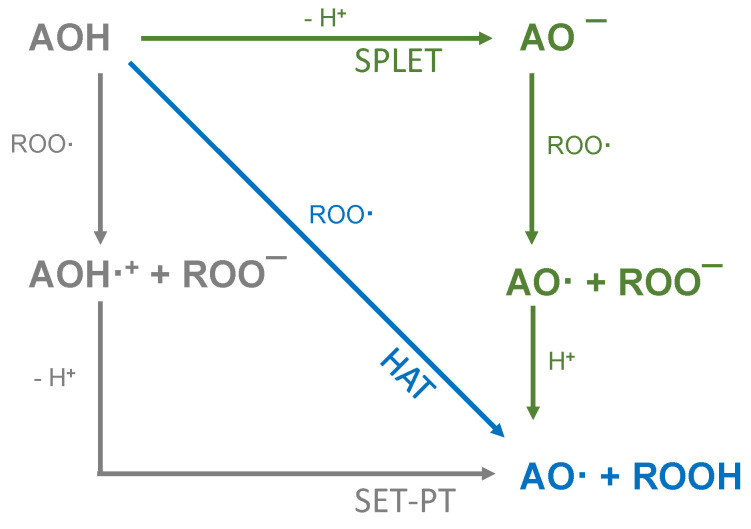
Schematic representation of HAT, single electron transfer followed by a proton transfer (SET-PT) and sequential proton loss electron transfer (SPLET) antioxidant reaction mechanisms (reproduced from Shang et al. (2009) [13] with some modifications).

**Figure 2 molecules-26-01244-f002:**
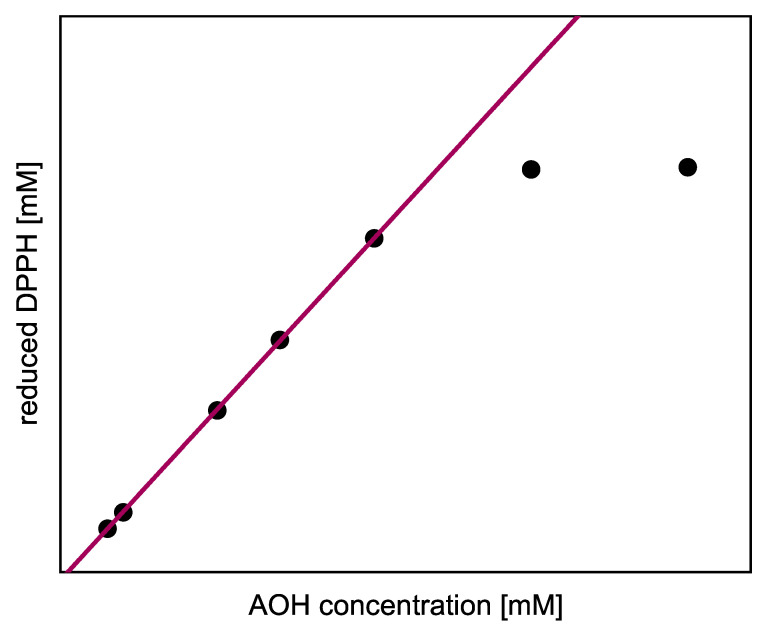
Schematic example of the linear regression for the evaluation of the total stoichiometry of all subreactions.

**Figure 3 molecules-26-01244-f003:**
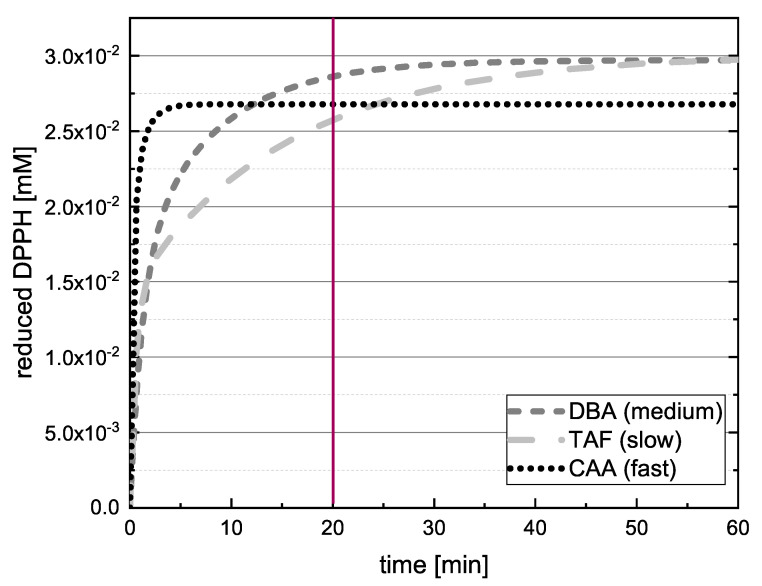
Representative fit curves showing the quantity of reduced DPPH radical ($\Delta {\left[\mathrm{DPPH}\cdot \right]}_{t}={\left[\mathrm{DPPH}\cdot \right]}_{0}-{\left[\mathrm{DPPH}\cdot \right]}_{t}$) in reactions with DBA, CAA and TAF as a function of time, demonstrating different kinetic behaviors.

**Figure 4 molecules-26-01244-f004:**
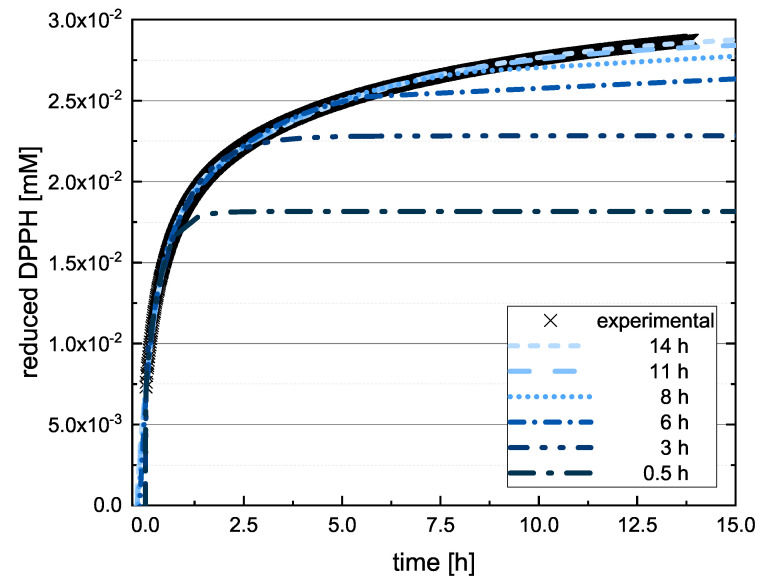
Representative measurement curve of the quantity of reduced DPPH radical in reactions with EPC as a function of time. Equation (12) is adapted for different time periods.

**Figure 5 molecules-26-01244-f005:**
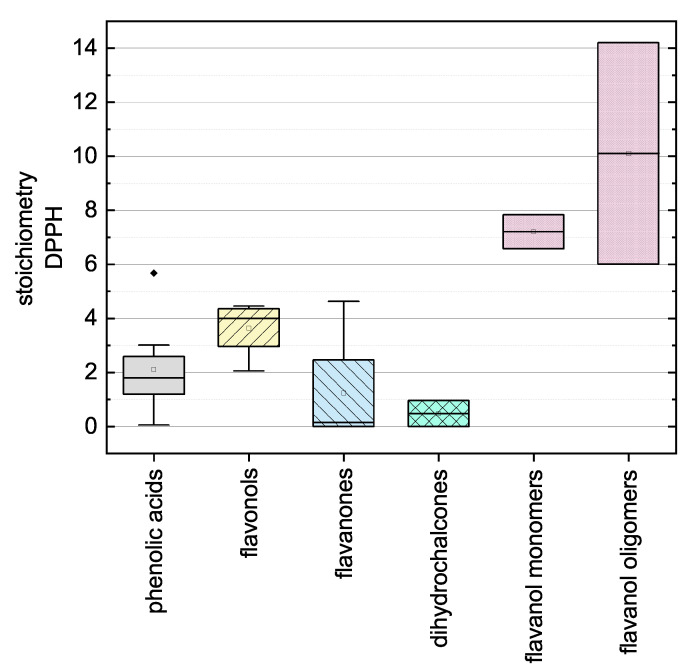
Boxplot of the mean stoichiometry values for subgroups of the phenolic compounds tested in the DPPH assay. Error bars represent range within 1.5 × IQR.

**Figure 6 molecules-26-01244-f006:**
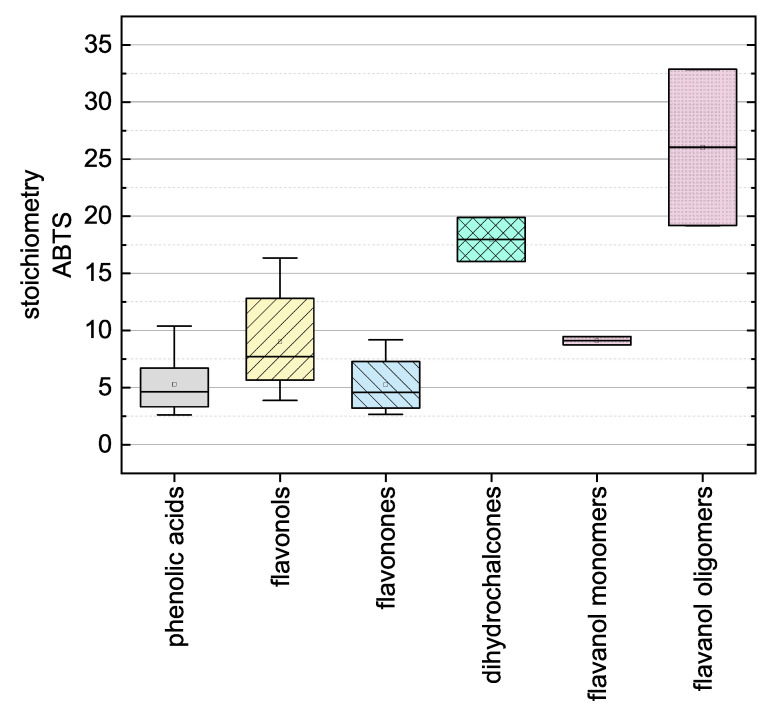
Boxplot of the mean stoichiometry values for subgroups of the phenolic compounds tested in the ABTS assay. Error bars represent range within 1.5 × IQR.

**Figure 7 molecules-26-01244-f007:**
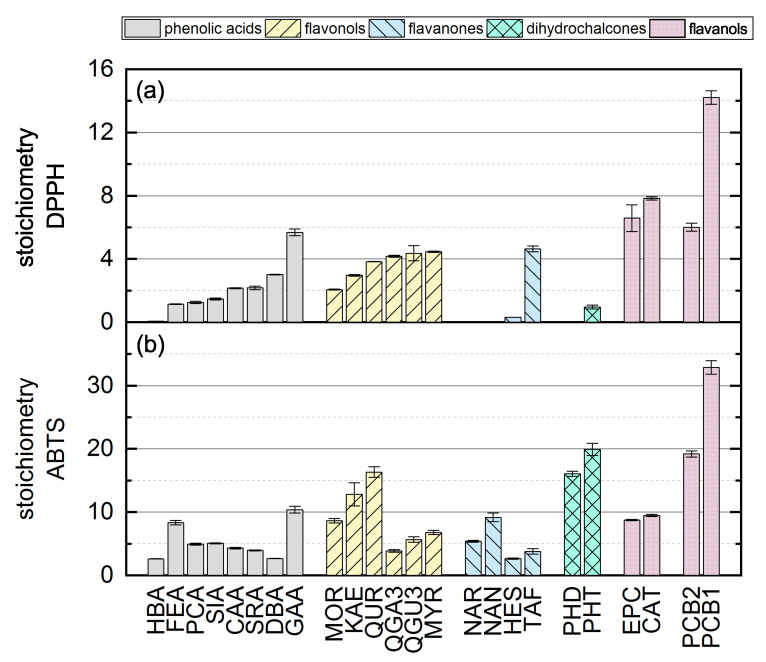
The antioxidant activity of all standard references in the (**a**) DPPH and (**b**) ABTS assays. Data are mean values of three measurements and standard errors.

**Table 1 molecules-26-01244-t001:** Summary of the phenolic compounds, reference standards, sample codes and corresponding side groups.

Group	Reference Standard	Sample Code	Side Group
**phenolic acids**		**1**	**3**	**4**	**5**			
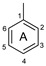	caffeic acids	CAA	(CH)_2_COOH	OH	OH	H			
3,4-dihydroxybenzoic acid	DBA	COOH	OH	OH	H			
ferulic acid	FEA	(CH)_2_COOH	OH	OCH_3_	H			
gallic acid	GAA	COOH	OH	OH	OH			
4-hydroxybenzoic acid	HBA	COOH	H	OH	H			
*p*-coumaric aicid	PCA	(CH)_2_COOH	H	OH	H			
sinapic acid	SIA	(CH)_2_COOH	OCH_3_	OH	OCH_3_			
siringic acid	SRA	COOH	OCH_3_	OH	OCH_3_			
**flavonols**		**2′**	**3′**	**4′**	**5′**	**3**	**5**	**7**
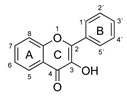	kaempferol	KAE	H	H	OH	H	OH	OH	OH
myricetin	MYR	H	OH	OH	OH	OH	OH	OH
morin	MOR	OH	H	OH	H	OH	OH	OH
quercetin-3-D-galactoside	QGA3	H	OH	OH	H	Glc	OH	OH
quercetin-3-D-glucoside	QGU3	H	OH	OH	H	Gal	OH	OH
quercetin	QUR	H	OH	OH	H	OH	OH	OH
**flavanones**		**3′**	**4′**	**5**	**7**			
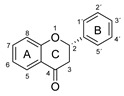	hesperetin	HES	OH	OCH_3_	OH	OH			
narirutin	NAR	H	OH	OH	2 Glc			
naringenin	NAN	H	OH	OH	OH			
taxifolin	TAF	OH	OH	OH	OH			
**dihydrochalcones**		**4**	**2′**	**4′**	**6′**			
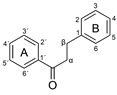	phloridzin	PHD	OH	OH	OH	Glc			
phloretin	PHT	OH	OH	OH	OH			
**flavanols**		**3′**	**4′**	**3**	**4**	**5**	**7**	
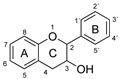	(+)-catechin	CAT	OH	OH	OH	H	OH	OH	
(−)-epicatechin	EPC	OH	OH	OH	H	OH	OH	
proanthocyanidin B1	PCB1	OH	OH	OH	CAT	OH	OH	
proanthocyanidin B2	PCB2	OH	OH	OH	EPC	OH	OH	

**Table 2 molecules-26-01244-t002:** Results of the DPPH assay presented as stoichiometry values after 5 and 30 min and stationary final values with literature comparisons and kinetic behaviors. Data are mean values of three measurements and standard errors. Sample names and structures are shown in Table 1.

Group	Sample Code	Stoichiometry (5 min)	Stoichiometry (30 min)	Stoichiometry (Steady Sate)	Kinetic Behavior	Stoichiometry (Literature)	Sources
phenolic acids	CAA	2.20 ± 0.06	2.24 ± 0.04	2.15 ± 0.03	fast	1.7–4.54	[18,23,27,42]
	DBA	2.51 ± 0.04	2.93 ± 0.11	3.01 ± 0.02	medium	2.1–2.8	[27,43,44]
	FEA	0.78 ± 0.02	1.11 ± 0.02	1.14 ± 0.01	medium	1.1–1.39 ${}^{d}$	[23,27,29,42]
	GAA	5.13 ± 0.26	5.71 ± 0.23	5.68 ± 0.21	fast	5.6–7.14 ${}^{d}$	[23,27,29]
	HBA	0.07 ± 0.00	0.06 ± 0.01	0.06 ± 0.01	fast	0	[35]
	PCA	0.14 ± 0.01	0.27 ± 0.03	1.25 ± 0.07	slow	0.9	[18]
	SIA	1.26 ± 0.04	1.31 ± 0.03	1.46 ± 0.06	medium	1.2 ${}^{d}$	[18]
	SRA	1.57 ± 0.10	2.12 ± 0.10	2.17 ± 0.10	medium	2.7	[27]
flavonols	KAE	2.96 ± 0.05	2.91 ± 0.05	2.96 ± 0.04	fast	1.8	[27]
	MOR	1.95 ± 0.02	2.05 ± 0.03	2.06 ± 0.03	medium	1.8	[27]
	MYR	3.26 ± 0.21	4.38 ± 0.03	4.46 ± 0.04	medium	7.6	[27]
	QGA3	2.33 ± 0.10	4.01 ± 0.09	4.17 ± 0.05	medium	-	
	QUR	2.27 ± 0.13	3.81 ± 0.02	3.83 ± 0.02	medium	4.86–5.2	[27,28]
	QGU3	3.23 ± 0.08	4.43 ± 0.27	4.36 ± 0.47	medium	3.78	[28]
flavonones	HES	0.12 ± 0.01	0.27 ± 0.01	0.30 ± 0.10	slow	0.9	[45]
	NAN	0	0	0	-	0	[35]
	NAR	0	0	0	-	-	
	TAF	2.89 ± 0.06	4.15 ± 0.09	4.63 ± 0.19	slow	4.18 ${}^{d}$	[46]
dihydrochalcones	PHD	0	0	0	-	-	
	PHT	0.16 ± 0.01	0.65 ± 0.03	0.96 ± 0.13	slow	-	
flavanols	CAT	2.39 ± 0.09	3.58 ± 0.05	7.84 ± 0.10	slow	3.72–4.5	[27,28]
	EPC	1.84 ± 0.01	2.51 ± 0.16	6.58 ± 0.84	slow	3.96–6.6	[27,28]
	PCB1	8.83 ± 0.13	12.29 ± 0.25	14.21 ± 0.42	slow	7.6	[27]
	PCB2	4.19 ± 0.36	5.81 ± 0.13	6.01 ± 0.26	medium	7.4	[27]

${}^{d}$ converted from EC_50_.

**Table 3 molecules-26-01244-t003:** Results of the ABTS assay presented as stoichiometry values after 5 and 30 min and stationary final values and kinetic behaviors. Data are mean values of three measurements and standard errors. Sample names and structures are shown in Table 1.

Group	Sample Code	Stoichiometry (5 min)	Stoichiometry (30 min)	Stoichiometry (Steady Sate)	Kinetic Behavior
phenolic acids	CAA	3.93 ± 0.14	4.35 ± 0.10	4.31 ± 0.13	medium
	DBA	2.60 ± 0.06	2.61 ± 0.06	2.66 ± 0.04	fast
	FEA	6.14 ± 0.25	8.46 ± 0.41	8.34 ± 0.37	medium
	GAA	8.47 ± 0.35	10.77 ± 0.50	10.38 ± 0.54	medium
	HBA	2.60 ± 0.02	2.59 ± 0.02	2.61 ± 0.02	fast
	PCA	4.58 ± 0.38	5.17 ± 0.23	4.95 ± 0.16	fast
	SIA	4.21 ± 0.06	5.04 ± 0.08	5.07 ± 0.08	medium
	SRA	3.98 ± 0.06	4.21 ± 0.05	3.96 ± 0.08	fast
flavonols	KAE	3.89 ± 0.46	5.33 ± 0.74	12.81 ± 1.82	slow
	MOR	8.64 ± 0.36	8.96 ± 0.50	7.45 ± 0.34	medium
	MYR	6.09 ± 0.16	6.86 ± 0.22	6.78 ± 0.18	medium
	QGA3	2.79 ± 0.09	3.69 ± 0.18	3.88 ± 0.20	medium
	QGU3	4.29 ± 0.19	5.84 ± 0.39	5.64 ± 0.43	medium
	QUR	9.67 ± 0.82	11.38 ± 0.28	12.42 ± 0.64	slow
flavonones	HES	2.93 ± 0.16	3.11 ± 0.11	2.66 ± 0.09	fast
	NAN	2.68 ± 0.21	4.63 ± 0.44	9.17 ± 0.69	slow
	NAR	0.69 ± 0.22	2.28 ± 0.34	5.39 ± 0.14	slow
	TAF	3.14 ± 0.12	3.76 ± 0.39	3.76 ± 0.43	fast
dihydrochalcones	PHD	5.20 ± 0.07	9.08 ± 0.06	16.05 ± 0.40	slow
	PHT	6.50 ± 0.29	12.56 ± 0.50	19.90 ± 0.96	slow
flavanols	CAT	9.30 ± 0.18	10.00 ± 0.26	9.47 ± 0.18	medium
	EPC	8.16 ± 0.05	8.78 ± 0.14	8.74 ± 0.12	medium
	PCB1	26.57 ± 0.41	31.48 ± 0.52	32.87 ± 1.06	medium
	PCB2	15.22 ± 0.23	18.61 ± 0.45	19.19 ± 0.50	medium

## Data Availability

Data of the measurement results are available from the authors.

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
