# Peer review of "Common Trends and Differences in Antioxidant Activity Analysis of Phenolic Substances Using Single Electron Transfer Based Assays"

_molecules, 2021, doi:10.3390/molecules26051244_

Round 1
Reviewer 1 Report
The work entitled: “Common trends and differences in antioxidant activity analysis of phenolic substances using single electron transfer based assays”, contributes to expanding knowledge about the ability of ABTS and DPPH methods for determining the antioxidant potential of phenolic compounds. This information will be very valuable to researchers using ABTS and DPPH to study potential antioxidant sources. The work is correctly carried out, particularly for the effect of pH, solvent and carbohydrate and protein content on the antioxidant activity of natural extracts.
I recommend revising the manuscript to avoid typing errors, such as on line 83 (“activity.We”)
Reviewer 2 Report
The authors present a comparative study of two of the usual methods for the evaluation of antioxidant activity. They apply it to 24 phenolic compounds from various families. It is well known that the different methods used to measure the antioxidant capacity produce very different results due to the different basis at the molecular level, different type of reaction or different accessibility of the reagents to the functional groups responsible for the antioxidant activity. All this has repercussions on differences in stoichiometry and kinetics. The work is interesting, well planned and with a detailed discussion of the results.
This reviewer has only encountered some difficulty, perhaps due to his own ignorance, in understanding some aspects of the methodology.
Line 120 shows the chemical equation (1) that corresponds to a HAT process, when previously (lines 41 and ss) it has been commented that reactions with ABTS and DPPH occur through SET or SPLET mechanisms.
In line 130 an equation (5) is presented that I think is not correct: the first term should be d[AO*] / dt
Reaction (3) is not considered in the kinetic equations. It is supposed to be unimportant, but the authors do not comment on it.
The second order kinetic equations are reformulated as pseudo-first order equations (7, 8,9), considering that the DPPH* concentration can be considered constant because it is in great excess. I believe that this is not the usual situation in these tests, since the parameter to be determined is the disappearance of DPPH *and if it were in a large excess, the reading would be not very sensitive.
I do not understand what the authors state in lines 146-148 and formula (11), which apparently is fundamental in the methodology used. For infinite time, the derivative of the increment of [DPPH] is obviously 0 (as follows from formula 11) since the kinetics will have reached its end.
From here I cannot understand how the authors apply this formula (11) to determine the stoichiometry at different times and at infinite time. Likewise, I do not understand what is represented in figure 3 and how it was obtained.
Figures 2 and 3 represent the y-axis “the quantity of DPPH reduced radical in reactions…”. Experimentally the test determines the disappearance of DPPH*, not the amount of DPPH2 formed; Equation (2) indicates that DPPH* is also consumed by binding to the antioxidant.
Reviewer 3 Report
For this paper, entitled ‘Common trends and differences in antioxidant activity analysis of phenolic substances using single electron transfer based assays’ fit the scope of the journal and seems to be interesting. However, concerns or some unclear important parts emerge in the paper that I will highlight to the authors to clarify:
- For our first discussions, Authors must give details or at least the key steps that led to analytical mathematical solutions of the 3 Consecutive antiradical kinetic system, the german reference 34 is not very accessible.
- More particularly I found ‘’t →∞ → α ̃k1[AOH]0 a Confusing result in the model, could you put the whole steps of those mathematics, this is important for the paper, in fact evaluations of endpoint measurements in the obtained equation 11 seems to be for a pseudo-zero-order reaction for the two reactants for instance, So you must transcribe a final equation 11 by writing clearly the two terms on either side of the symbol = so that I can understand.
- Also In the mathematical model, the biexponential equation 11 is decreasing, I end up with increasing experimental curves see material and methods, Need more details, does the y-axis represent the reduced DPPH concentration or the percentage of DPPH scavenging at the different time?
- from line 183 to 185, we need also more details about the slope expression?
- Although these studies made it possible to establish a ranking according to the order of antioxidant efficacy, you confirms the literature in the case of DPPH for some polyphenolics with the exception of dihydrochalcones class using a box-plots, but with the compared literature, they used only classical statistics, how could you justify this comparison? You need to show the advantage of the use of boxplots
- Also as minor comments:
Line 17,19 correct by: plant-based diet,
line 34 Quercetin does not have a pyrogallol moiety in B cycle, please rephrase
table 1, correct the name of caffeic acid,
the numbering of dihydrochalcones in table 1 is not appropriate please correct
Line 149 you say the measured absorption values, give wavelenght,
I really recommend to check all references given in the text not in the reference section, they do not correspond to what you quote, example reference 35 in line 250 where all compounds activity were measured in Ethanol, not in Methanol
Round 2
Reviewer 3 Report
It is necessary to show in the text the benefits to report the difference DPPH0 - DPPH. instead of DPPH. alone as a function of time
The approach of the mathematical solution of the model is fine now but still needs a minimum details as I have asked for in the first review, in fact the first solution of the equation 7 is easy to find, it is a first order equation, this must be replaced in the equation 8 wich will generate a first order differential equation with second member in the form of an exponential, a finding of a particular solution would be the key step to help determine Equation 10 solution which is the final step for the calculation of the concentration of DPPH. , could you discuss those steps without dictating solution 10 directly in the manuscript…. This is what I asked for but I didn't have an answer, also keep reference 33 in German, Give only page numbers in section references
By adding your antioxidant reagents, did you not observe problems with the absorbances obtained, saturation problems with radicals?
line 277 radicals
